# One-Step Construction of Tryptophan-Derived Small Molecule Hydrogels for Antibacterial Materials

**DOI:** 10.3390/molecules28083334

**Published:** 2023-04-10

**Authors:** Xianwen Song, Shunmei He, Jun Zheng, Shutong Yang, Qiang Li, Yi Zhang

**Affiliations:** Hunan Provincial Key Laboratory of Micro & Nano Materials Interface Science, College of Chemistry and Chemical Engineering, Central South University, Changsha 410083, China

**Keywords:** antimicrobial materials, amino acid-based hydrogels, tryptophan, self-assembly, hydrogen bonding

## Abstract

Amino acid-based hydrogels have received widespread attention because of their wide range of sources, biodegradability, and biocompatibility. Despite considerable progress, the development of such hydrogels has been limited by critical problems such as bacterial infection and complex preparation. Herein, by using the non-toxic gluconolactone (GDL) to adjust the pH of the solution to induce the rapid self-assembly of N-[(benzyloxy)carbonyl]-L-tryptophan (ZW) to form a three-dimensional (3D) gel network, we developed a stable and effective self-assembled small-molecule hydrogel. Characterization assays and molecular dynamics studies indicate that π–π stacking and hydrogen bonding are the main drivers of self-assembly between ZW molecules. In vitro experiments further confirmed this material’s sustained release properties, low cytotoxicity, and excellent antibacterial activity, particularly against Gram-negative *Escherichia coli* and Gram-positive *Staphylococcus aureus*. This study provides a different and innovative perspective for the further development of antibacterial materials based on amino acid derivatives.

## 1. Introduction

Recent studies have shown that amino acid-based hydrogels have great potential for various biotechnology applications, such as scaffolds for biomaterials, different types of drug delivery, and wound healing, due to their excellent biocompatibility and biodegradability [1,2,3]. Notably, the fibrous network of hydrogels has a large number of empty spaces that can trap large amounts of water molecules, which can provide cells with an optimally moist environment that is similar to physiological conditions [4,5]. However, the moist environment also provides ideal conditions for bacterial growth, making the hydrogel susceptible to outside microbial infections when in use. Such infections would cause much inconvenience in many practical applications, especially in tissue engineering and cell culture. Therefore, much work has focused on designing and developing hydrogels with antimicrobial properties [6,7,8]. In addition to adding antimicrobial agents directly, methods of developing antimicrobial hydrogels include the design of amphiphilic polymers, the addition of metal ions, and the synthesis of antimicrobial peptides [8,9,10]. For example, Li et al. synthesized a series of star-shaped and membrane-active cationic polyether amines through the condensation of amino acids and polyether amines, which can disrupt bacterial membranes and lead to the leakage of the internal cytoplasm. These materials may introduce certain toxic chemicals during the complex synthesis and purification process, leading to various side effects such as inflammation, immunogenicity, and oxidative DNA damage, ultimately causing unavoidable damage to normal cells [11,12,13]. The preparation of simple and effective antibacterial hydrogels without any chemical modifications or other molecules remains a pressing issue in the development of amino acid derivatives.

Tryptophan has been extensively studied due to its numerous advantages, including its low price, its wide sources, its excellent biocompatibility, and its bioactivity [14,15,16]. However, the nature of its poor water solubility, its single function, and its low bioavailability continue to hinder its further applications. Through the synthesis of tryptophan derivatives and self-assembly, the obtained hydrogel not only maintains the inherent biocompatibility of amino acids but also has high water content, diversity, and high utilization [17,18,19]. In addition, multiple modification options provide many possibilities for the material properties and biological functions of such a hydrogel, which is desirable in fabricating antibacterial hydrogels with excellent properties and expanding the application of small molecule hydrogel materials in biomedicine [11,20]. 

Regarding tryptophan and its derivatives, several tryptophan-rich marine alkaloids reported over the years have exhibited strong activity against bacteria, viruses, or fungi. In addition, a variety of antimicrobial peptides consisting of a high frequency of tryptophan, such as tritrpticin and indolicidin, have been suggested to play an important role in the antimicrobial properties of this polypeptide [21,22,23].

In this work, we report a tryptophan-derivative hydrogel, which is self-assembled by N-[(phenyl methoxy)carbonyl]-L-tryptophan (ZW, where Z represents the benzene ring and W represents tryptophan) through intermolecular interactions. Under the modulation of gluconolactone (GDL), the sol-gel process of ZW solutions can be successfully achieved in a certain pH range. In most cases, ZW is used as a raw material for synthesis and as an intermediate in medicine; therefore, its properties have not been fully reported [24]. 

Spectral characterization and kinetic simulations showed that the gelation process can be attributed to two things: π–π interactions provided by the indole and benzene rings and hydrogen bonding provided by the amide and carboxyl groups. When delivered as a hydrogel, ZW possesses better biocompatibility and self-releasing behavior. Even better, the ZW hydrogel can be prepared in a few minutes, unlike the time-consuming process of other materials. In vitro experiments have also demonstrated the excellent antibacterial activity of the hydrogel, including its activity against Gram-negative *Escherichia coli* and Gram-positive *Staphylococcus aureus*. Therefore, given its excellent biocompatibility and antimicrobial properties, the hydrogel reported in this study has great potential as a broad-spectrum antimicrobial agent in treating wounds and skin infections [25,26,27]. In addition, this material has the potential for future clinical applications due to its ease of preparation and the non-toxic and non-hazardous nature of the raw materials.

## 2. Results and Discussion

### 2.1. Hydrogel Preparation

As a general procedure, 30.0 mg ZW powder was dissolved in 5 mL transparent NaHCO_3_ solution (pH 8.3) through sonication to obtain a clear and transparent solution. Then, 0.1 mol/L gluconolactone (GDL) was added to the solution at 37 °C and the tube was inverted to confirm gelation. The GDL is non-toxic and biocompatible and will gradually break down and release gluconic acid to change the pH of the solution [28,29]. When the pH reached 7.5, the ZW solution transformed into a self-supporting hydrogel after 2 min. When the pH was adjusted to 5.0–6.5, this sol-to-gel phase transition occurred rapidly, within 1 min (Figure 1A). It is worth noting that as the pH changed from 8.33 to 5.0, the ZW solution changed from a transparent to a translucent state and, eventually, to a white and completely opaque hydrogel. We refer to the material obtained by the above method as a ZW/GDL hydrogel, although of course it can also be successfully prepared using other acid agents (e.g., hydrochloric acid).

The morphology of the ZW/GDL hydrogels was characterized by scanning electron microscopy (SEM) and the results are shown in Figure 1B. SEM demonstrated that the diluted gels exhibited a filamentary structure consisting of nanofibers with diameters of approximately 40–60 nm. These morphological features all indicated that multiple fibers with lengths greater than 600 nm were entangled with each other to produce a three-dimensional network structure, i.e., a macroscopic gel [30,31]. It was also observed that there was a dense interweaving of nanofibers in these undiluted three-dimensional networks, particularly in hydrogels at pH 5.0. These results suggest that the formation of multiple intersecting fibers by the ZW molecules is important in producing the sol-gel transition.

In order to determine the mechanical properties of ZW/GDL hydrogels with different three-dimensional networks, a rheological study was carried out. According to the dynamic strain scan (frequency = 1 rad/s), the storage modulus (G′) values for the hydrogels at different pH were favored over the loss modulus (G″) in the linear viscoelastic region (Figure 1C) [32,33]. The critical strain values for the gels were calculated from the graph to be approximately 16% to 23%, which indicated relatively strong gel formation. When the strain exceeded 23%, the G″ was higher than the G′, indicating a transition from the gel state to the solution state. As can be seen from the dynamic frequency scan (Figure 1D), all hydrogels showed weak frequency-dependent behavior, with the G′ higher than the G″ for each gel in the region of 0.1–100 rad/s, indicating the solid character of these hydrogels. Among all the groups, the ZW/GDL hydrogels had the lowest modulus at pH 6.5. Furthermore, when the concentration of ZW was fixed at 6 mg/mL, the mechanical strength of the ZW/GDL hydrogels increased with decreasing pH, but there was a maximum at one point. When the pH was reduced from 6.5 to 6.0, the G′ value doubled significantly, from 463 Pa to 950 Pa. In contrast, when the pH was reduced from 6.0 to 5.5, there was almost no change in the G′, indicating that the modulus had gradually reached its upper limit. 

In addition, rheological studies of the ZW/GDL hydrogels with different concentrations were investigated. As shown in Appendix A, the mechanical strength of the hydrogels tended to increase with an increasing ZW concentration, mainly due to more nanofibers being entangled. Furthermore, in successive step-strain measurements (Appendix A), the G″ was higher than the G′ at a high-magnitude strain of 300%, while the G′ was greater than the G″ at a low-magnitude strain of 0.1%, indicating that the sol reverted to a gel [34]. This phenomenon could be repeated many times, showing that the resulting hydrogel has good shear thinning and self-healing properties. These results reflect the excellent mechanical strength of the ZW/GDL hydrogels and indicate their potential for biomedical applications.

### 2.2. Dynamics Simulation

To simulate the possible gelation pathway of the selected small molecules in solution, we first used all-atom molecular dynamics simulations (AAMD) to make predictions at the microscopic scale [35]. Figure 2A shows the optimized structure of the ZW molecule. The dynamic processes and interactions between small molecules were mainly studied. Initially, 50 molecules were randomly contained in a cube of length 50 Å filled with the aqueous solution and equilibrated at 298.15 K for 1 ns. Finally, MD simulations of 50 ns were performed. Eventually, all molecules formed the final aggregated structure. We extracted the molecular dynamics simulations at 0 ns, 0.5 ns, 10 ns, and 50 ns, respectively, and concealed the water molecules for convenience. It can be seen from the simulation results that the ZW molecules very easily formed aggregates. At 10 ns, most of the ZW molecules were already concentrated in one region. Finally, all the molecules clearly formed aggregates at 50 ns.

From a certain set of structures extracted from the final simulation, it can be seen that the driving force for the formation of aggregates between the ZW molecules was mainly due to the π–π stacking of the benzene ring and the hydrogen bonding that existed between the carboxylic acid and the amide (Figure 2B). Note that this driving force, while prone to precipitation of molecular aggregates, may also contribute to the formation of fibers.

### 2.3. Self-Assembly Mechanism of Hydrogel

The driving force for the aggregation of ZW molecules was investigated by 1H NMR. Due to its high modulus, the ZW/GDL hydrogel at pH 6.0 was used for all tests. Multiple proton signals of the hydrogel showed significant upfield shifts (Figure 3A). For example, protons belonging to the indole ring were shifted from 7.07, 6.98, 7.55, and 7.16 to 7.01, 6.88, 7.49, and 7.04, respectively, while protons belonging to the benzene ring were shifted from 7.36 to 7.30 (ppm). According to previous reports, an aromatic proton enters the shielding region of the π-ring current-induced magnetic field of another aromatic ring during the π–π stacking process, which weakens the de-shielding of the aromatic proton and, thus, leads to an up-field transfer during the assembly process [36,37]. These results indicate that strong π–π stacking occurs in both the indole and benzene rings.

The FT-IR spectroscopy test was performed on the samples before and after gelation to investigate the ZW/GDL hydrogel formation. As shown in Figure 3B, a new single sharp peak at 1589 cm^−1^ suggested that ordered π–π stacking occurred in the formation of a hydrogel. In the 3500–3300 cm^−1^ range, the original O-H/N-H stretching vibration spike (3386 cm^−1^ and 3421 cm^−1^) shifted to a distinct broad peak, indicating the presence of intermolecular hydrogen bonds during the self-assembly from the solution to the hydrogel [35,38]. Moreover, The C=O stretching vibration peak related to the ester group at 1731 cm^−1^ disappeared and the vibration associated with the carboxyl group at 1698 cm^−1^ was enhanced after self-assembly, indicating that the carboxyl and ester groups of the ZW molecule were involved in the formation of the hydrogen bonds. Therefore, it was clear that hydrogen bonding was one of the primary driving forces for gelation.

To further verify these self-assembly drivers, ultraviolet-visible absorption spectroscopy (UV-Vis) tests were carried out. As shown in Figure 3C, the absorption peak of the ZW molecule in solution was 215 nm, while in the hydrogel it shifted to 220 nm. In addition, the hydrogel showed a slow shift (close to 222 nm) as the concentration increased (Appendix A). This apparent red-shift phenomenon was the result of enhanced π-π interactions, suggesting that intermolecular π–π interactions are an important driving force for the gelation of ZW and that the polymerization mode undergoes a shift from H-type to J-type [39,40].

In addition, the circular dichroism (CD) spectrum was used to explore the chiral character of ZW in the hydrogel and the solution (Figure 3D). Both showed a positive Cotton effect peak at 225–240 nm, suggesting that the presence of right-handed helical assemblies and the stacking mode of ZW molecules was not changed after assembly [41]. Compared with the solution, the hydrogel exhibited an additional weak positive peak near 280 nm and a significant red-shift of the maximum CD peak, which was consistent with the results of the UV-vis spectra and further proved that the J-aggregation of the ZW molecules occurred during the self-assembly process.

### 2.4. Drug Release and Cytotoxicity Evaluation

The sustained drug-release capacity was evaluated (Appendix A). The accumulative drug release rate of the hydrogel was monitored over time under a simulated normal physiological condition (PBS, pH 7.4). As shown in Figure 4A, the initial rapid drug release of nearly 30% was recorded within 12 h, followed by a gradual release process. Finally, the accumulative release rate reached 32% in 48 h. The sustained release ability of the hydrogel could be attributed to the solubility and the brittleness of the fiber networks [42], which greatly improved its bioavailability and promoted its potential biomedical application.

It is well known that good biocompatibility is an important requirement for materials that are used in many biomedical applications [43,44,45]. Therefore, the CCK-8 assay was performed to assess and analyze the cytotoxicity of the hydrogels at different ZW concentrations. After co-incubation of ZW/GDL hydrogels with L929 fibroblasts for 24 h, cell viability was assessed using CCK-8 reagents. The results are shown in Figure 4B. Compared to the control, the hydrogels did not induce significant cytotoxic effects in the ZW concentration range of 4–10 mg/L. In contrast, the cell viability of the hydrogel decreased to 82% at 12 mg/mL, indicating that the excessive drug content was detrimental to cell growth. These results indicate that this hydrogel material has high biocompatibility and provides an important basis for its biological study.

### 2.5. Antibacterial Properties

The polar amide groups and the hydrophobic ring structures on the ZW molecular structure facilitate its localization at polar or nonpolar interfaces. It may interact with the hydrophobic part of the bacterial cell membrane through its aromatic group, thereby disrupting the membrane structure to exert its antibacterial effect, as used by antibacterial peptides. In addition, the nanofibers network has good brittleness, enabling slow drug release, which continuously exerts the antibacterial ability of ZW. Therefore, we investigated the antibacterial ability of this tryptophan derivative hydrogel [46,47].

Bacterial infections are now considered to be the most common type of infection causing mortality and morbidity in humans, as well as being the leading cause of wound infections. Among bacteria, *Escherichia coli* (*E. coli*) and *Staphylococcus aureus* (*S. aureus*) are the most common pathogens [48,49,50]. Accordingly, to assess the antimicrobial activity of the ZW/GDL hydrogel, we chose these two bacteria as experimental models. Bacterial proliferation was evaluated by measuring the optical density (OD) at 600 nm. As shown in the time-dependent antimicrobial test of the hydrogel, for *E. coli* (Figure 5A), the hydrogels showed excellent antibacterial efficacy, while the ZW solution could not effectively inhibit bacterial growth and reproduction. For *S. aureus* (Figure 5B), the inhibition rate of ZW/GDL hydrogel reached 90% and showed a robust antimicrobial effect within 12 h. In contrast, the bacteria in the solution showed the same rapid reproduction as that of the control group. This could be also demonstrated by the agar plate test results.

The hydrogels were incubated with different concentrations of *E. coli* and *S. aureus* for 48 h to evaluate their antimicrobial capacity. Compared with the blank group, no bacterial growth was evident on agar plates treated with hydrogel, but the agar plates treated with the ZW solution showed a large number of colonies (Figure 5C,D). This also indicated that the hydrogel could completely inhibit bacteria below the concentration of 1 × 10^8^ colony-forming units (CFU)/mL, but the antimicrobial ability would fade when subjected to 1 × 10^8^ CFU/mL bacteria (Appendix A). These results indicated that ZW/GDL hydrogel had outstanding antibacterial effects against both *E. coli* (gram-negative) and *S. aureus* (gram-positive).

The hydrogel exhibited an excellent antibacterial effect against both bacteria, which was superior to the effect of the solution. This effect of the hydrogel may be related to its nature, as the antibacterial effect of ionized ZW was not obvious. This simple supramolecular hydrogel is expected to be further applied in the field of wound healing and to provide more options for the future development of broad-spectrum antibacterial drugs to treat diseases caused by bacterial infections.

## 3. Materials and Methods

### 3.1. Materials

N-[(phenyl methoxy) carbonyl]-L-tryptophan (ZW) and gluconolactone (GDL) were purchased from GL Biochem Co., Ltd. (Shanghai, China). Sodium bicarbonate (NaHCO_3_), phosphate buffer solutions (PBS), and fetal bovine serum (FBS) were purchased from Cusabio Biotech Co., Ltd. (Wuhan, China), Ltd. RPMI-1640 Medium, Trypsin-EDTA, and penicillin-streptomycin were obtained from Solarbio (Beijing, China). The L-929 (mouse fibro cell line) was purchased from Fenghbio Science and Technology Ltd. (Changsha, China). *Escherichia coli* (*E. coli*) and *Staphylococcus aureus* (*S. aureus*) were obtained from Luwei Technology Co. Ltd. (Shanghai China). Luria–Bertani (LB) broth and LB agar were acquired from Hangzhou Baisi Bio-tech Co. Ltd. All of the above reagents were used directly without further purification, and ultrapure water was used throughout this study.

### 3.2. Preparation of Hydrogels

A saturated NaHCO_3_ (pH 8.3, 0.1 mol/L) solution was first prepared, then ZW powder was added and sonicated to obtain a homogeneous and clear 6 mg/mL ZW solution. The resulting 5 mL ZW solution was then adjusted to pH 7.0 with a 0.1 mol/L GDL solution and mixed by rapid shaking. The pH of the ZW system was then adjusted to 6.0–6.5. A homogeneous stable hydrogel was obtained after standing at room temperature for 1 min with a minimum gel concentration of 4.0 mg/mL.

### 3.3. Characterization

The ZW solution and ZW/GDL hydrogel were fully freeze-dried and then dissolved in DMSO-d6. The samples were scanned using an AMX-400 (Bruker, Switzerland) at 300 K. The 1H NMR spectra of the samples were recorded. ZW/GDL hydrogels of different pH values were freeze-dried; then, their microstructure images were characterized by scanning electron microscopy (SEM, JEOL JSM-6701F, Japan). The vacuum-dried lyophilized hydrogel powders were mixed and ground with KBr crystals and the lyophilized baked under IR light for 1 h. FT-IR spectra of the samples were collected by the Perkin Elmer Spectrum One instrument (USA) in the scan range of 4000–400 cm^−1^.

### 3.4. All-Atom Molecular Dynamics (AAMD)

The small molecule ZW was built by Gaussian View 16 and geometrically optimized by the B3LYP method and the 6-311G (d, p) basis set. Afterward, the electrostatic potential (ESP) was calculated by Guassian16 using the HF/6-31G* method and the basis set. Molecular dynamics simulations were performed in the GROMACS (version 2020.6) simulation package, using the generalized amber force field (GAFF) in conjunction with the TIP3P water model. Initially, 50 molecules were packed randomly in a cube of 50 Å length using GROMACS software. After dissolution with the water molecules, the solution was minimized using a conjugate gradient algorithm with a maximum force tolerance of 200 KJ/mol. Then, the temperature and the volume of each system were equilibrated by running constant volume (NVT) for 100 ps, followed by a constant temperature simulation by running constant pressure (NPT) for 100 ps. A production run of 50 ns was then run in the NPT setting. This process used the LINCS algorithm to constrain all covalent bonds to the hydrogen atoms. The final simulation results are displayed by the VMD 1.9.3 software.

### 3.5. Rheological Mechanical

The rheological properties of ZW/GDL hydrogels were assessed using a rotational rheometer (Anton Paar, MCR 302) at 25°, with a parallel plate of 20 mm diameter employed at a gap of 1 mm. First, changes in storage modulus (G′) and loss modulus (G″) were recorded in the region of 1–100 rad/s by means of a dynamic frequency sweep mode with 0.1% strain. Next, dynamic strain sweep experiments were carried out at 1 rad/s in the region of 0.1–100% of the strain. Finally, an alternate-step strain sweep test was performed at a constant frequency (1 rad/s) and the strain was varied from 0.1% to 300%. The entire test was repeated three times and kept at 25 °C.

### 3.6. Sustained Release Test

UV-Vis spectra of the sample were measured by a Shimadzu UV-2450 spectrometer (Japan). Approximately 1 mL sample was scanned from 200 nm to 400 nm in a 1 cm quartz cell.

Hydrogel (1 mL, 6 mg/mL) was prepared in a centrifuge tube containing 0.4 mL PBS buffer solution (pH 7.4, 0.01 M) and continually oscillated in a thermostatic water bath of 37 °C. At each time point, 200 μL of the release solution was removed and supplemented with an equal amount of fresh PBS solution. The ZW quantity of the release solution was determined by measuring the UV absorbance of 215 nm. Each experiment was performed three times.

### 3.7. Cytotoxicity Assay

The L929 cells (mouse fibroblast cell line) were cultured using RPMI-1640 at 37 °C in a humidified environment with 5% CO_2_. The medium was supplemented with 10% (*v*/*v*) FBS and 1% penicillin-streptomycin. Passaging was ensured every other day during the culture. For cytotoxicity experiments, the L929 cells were housed in 0.1 mL medium in 96-well plates (6 × 10^4^ cells/mL) and incubated for 24 h. After replacing the serum-free medium for 24 h, the incubation was continued for 24 h by adding ZW/GDL hydrogels with different concentrations. Then, the old culture medium was removed and a 0.1 mL culture medium containing 10% (*v*/*v*) CCK-8 reagent was added to each of the 96 wells and incubated with the cells for 2.5 h in a CO_2_ incubator. Finally, the absorbance of each well at 450 nm was measured using a TU-1901 microtiter reader (Molecular Devices, Massachusetts, USA).

The cell viability was based on the following formula: cell viability (%) = [(OD _treated_ − OD _blank_)/(OD _control_ − OD _blank_) × 100%].

### 3.8. Antimicrobial Activity Studies

Bacteria of *Escherichia coli* (*E. coli*, ATCC 25922) strains and *Staphylococcus aureus* (*S. aureus*, ATCC 25923) were grown in Luria–Bertani broth medium (LB) by a shake flask method at 37 °C. The bacteria with a concentration of 1 × 10^8^ CFU/mL were obtained by measuring the absorbance at 600 nm and by the counting method. A 2 mL bacteria solution diluted to 1 × 10^6^ CFU/mL with sterile LB broth was transferred to a sterile centrifuge tube containing 500 μL of sample and oscillated in a thermostatic shaker (37 °C, 120 r/min) for 4 h. Then, 50 μL bacterial solution was diluted and spread evenly on the agar plate and the antibacterial effects were evaluated by cultivating in a 37 °C incubator for 12 h. For the time-dependent antibacterial assay, the centrifuge tube containing 2 mL of bacteria solution (1 × 10^6^ CFU/mL) and 500 μL of ZW/GDL hydrogel was cultured in a 37 °C shaker, and 100 uL of bacteria solution was removed every 2 h to monitor the OD600 reading. The bacteria solution without hydrogel was used as a control. For concentration-dependent antibacterial experiments, 2 mL of bacteria solution with three different bacteria concentrations (1 × 10^6^ CFU/mL, 1 × 10^7^ CFU/mL, 1 × 10^8^ CFU/mL) was added to the centrifuge tube containing 500 μL of ZW/GDL hydrogel (6 mg/mL). Then, 100 uL of bacteria solution was removed after 48 h incubation to record the OD_600_ value.

## 4. Conclusions

In this study, we developed a facile tryptophan derivative/gluconolactone (ZW/GDL) small-molecule hydrogel as a rapidly preparable antimicrobial material. By adjusting the pH through GDL, small ZW molecules in solution can rapidly self-assemble into nanofibrous hydrogels through hydrogen bonding interactions and π–π stacking. ZW/GDL hydrogels also exhibit good drug retardation properties and biocompatibility with normal mammalian cells. Moreover, the in vivo experiments demonstrated that ZW hydrogels exhibited significant antibacterial activity, compared to that of free ZW, such as Gram-negative *Escherichia coli* and Gram-positive *Staphylococcus aureus*. Because of these advantages, this hydrogel is highly likely to be commercialized, providing a different paradigm for designing and developing antimicrobial hydrogel materials based on amino acid derivatives.

## Figures and Tables

**Figure 1 molecules-28-03334-f001:**
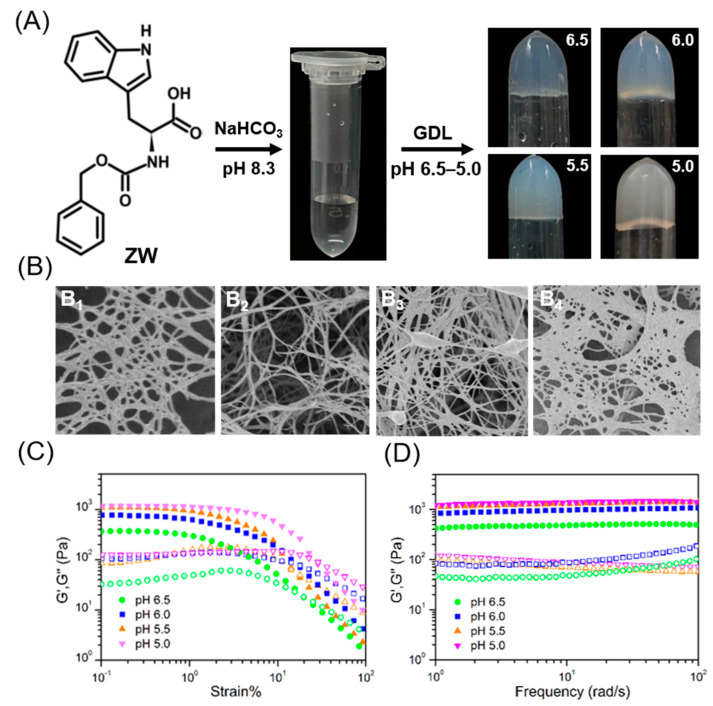
(**A**) Demonstration of the facile preparation of ZW/GDL hydrogels at different pH. (**B**) B1, B2, B3, and B4 refer to SEM images obtained at pH 5.0, 5.5, 6.0, and 6.5, respectively (scale bar: 100 nm). (**C**) strain-dependent sweep, and (**D**) dynamic frequency sweep of hydrogels at pH 6.5, pH 6.0, pH 6.0, and pH 5.0.

**Figure 2 molecules-28-03334-f002:**
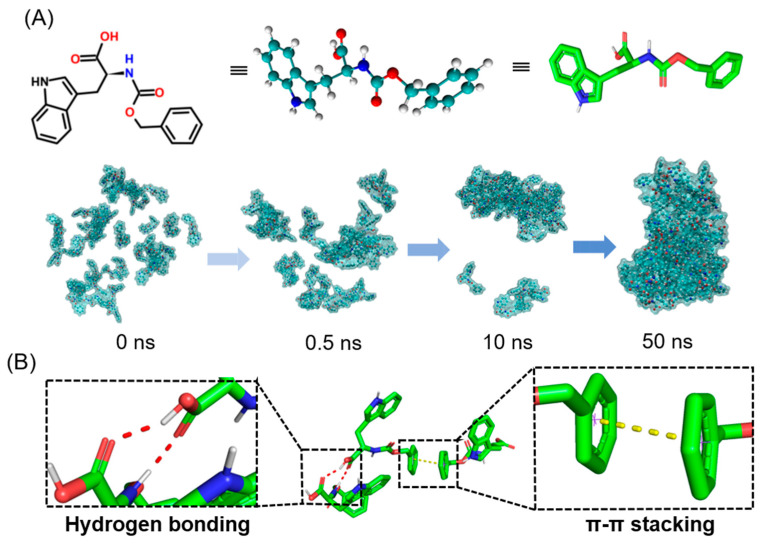
(**A**) Molecular dynamics simulation in water environment within 50 ns. (**B**) Hydrogen bonding and π–π stacking between ZW molecules.

**Figure 3 molecules-28-03334-f003:**
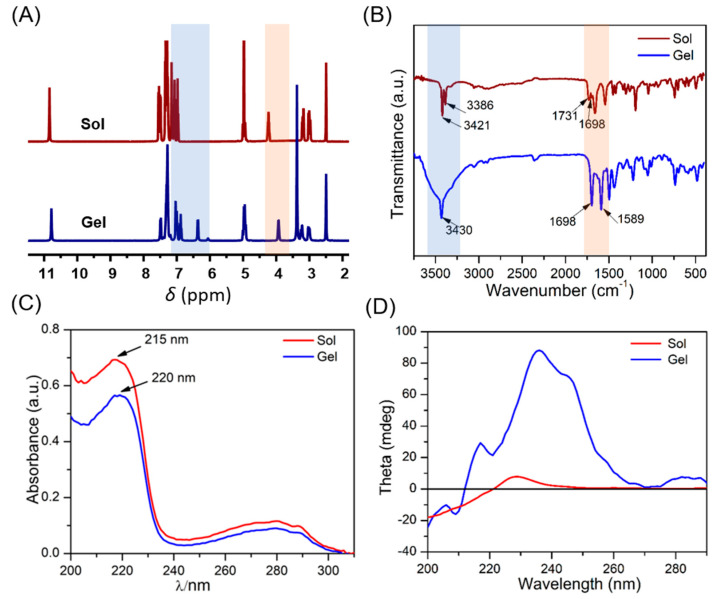
(**A**) 1H NMR spectra, (**B**) FT-IR spectra, (**C**) UV-Vis spectra, and (**D**) CD of ZW solution and ZW/GDL hydrogel (pH 6.0).

**Figure 4 molecules-28-03334-f004:**
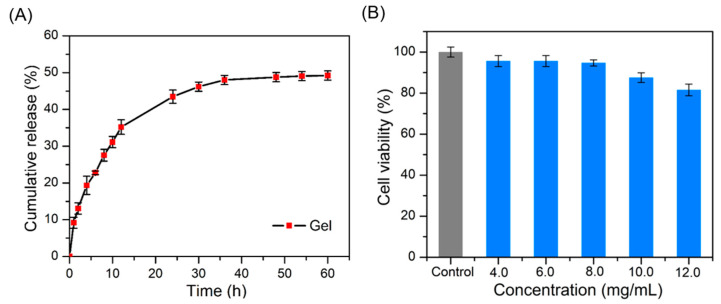
(**A**) In vitro release profile from ZW/GDL hydrogel incubated with PBS solution (pH 7.4) at 37 °C. (**B**) Cell viability of L929 fibroblasts incubated with different concentrations of hydrogel for 24 h.

**Figure 5 molecules-28-03334-f005:**
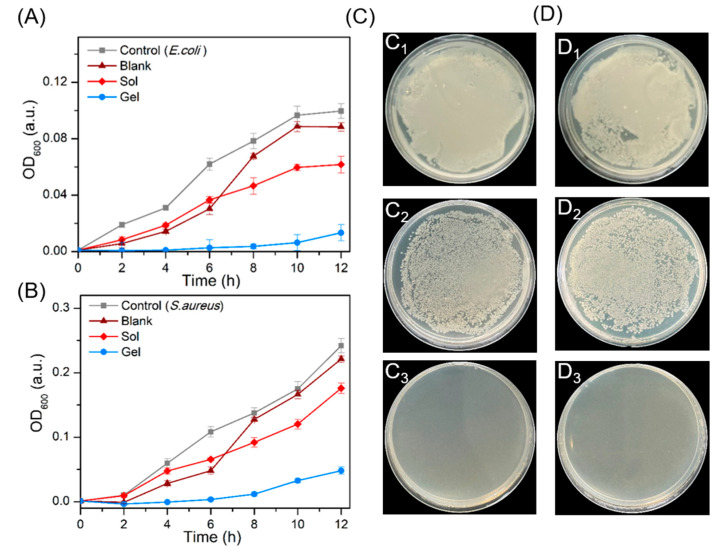
Time-dependent antimicrobial effects of the ZW solution and ZW/GDL hydrogel toward (**A**) *E. coli* and (**B**) *S. aureus* bacteria. (**C**) Photographs of bacterial colonies formed by *E. coli* treated with control (**C_1_**), solution (**C_2_**), and hydrogel (**C_3_**). (**D**) Photographs of bacterial colonies formed by *S. aureus* treated with control (**D_1_**), solution (**D_2_**), and hydrogel (**D_3_**). For comparison, blank refers to NAHCO_3_/GDL solution of pH 6.0 without ZW.

## Data Availability

Not applicable.

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
