# Peer review of "One-Step Construction of Tryptophan-Derived Small Molecule Hydrogels for Antibacterial Materials"

_molecules, 2023, doi:10.3390/molecules28083334_

Round 1

Reviewer 1 Report

Comment for molecules-2269413-peer-review-v1

1.      The manuscript reports a hydrogel synthesis that is applicable for various purposes. The synthesis of tryptophan-containing hydrogels is not a new material, but one of the advantages of this hydrogel that the authors focused on is its ability to inhibit bacterial growth. On the other hand, the manuscript has also been well prepared. However, from a literature search it can be seen that this work was also reported in 2021

(https://doi.org/10.1016/j.cclet.2022.108069) but unfortunately it was not cited in this manuscript. Is it intentional or does it need explanation? Also, the author needs to explain the significance of the difference in data and the significant improvement from papers that have been published and those that are currently submitted.

2.      Rules for writing scientific names of microbial isolates to be considered.

3.      It has been said that the synthesized hydrogels have great potential for the treatment of skin or wound infections caused by Gram positive and Gram negative bacteria. This conclusion must be supported by data showing that the antibacterial effect of the hydrogel has a broad antibacterial spectrum. If the data already exists, it should be mentioned in this manuscript.

4.      It is advisable that the introductory article of this manuscript is also cited.

Author Response

Point-by-point response to the individual questions of Reviewer 1:

Over all Comment:

Comment 1: proceed to an extensive editing of English language, of their text. The manuscript reports a hydrogel synthesis that is applicable for various purposes. The synthesis of tryptophan-containing hydrogels is not a new material, but one of the advantages of this hydrogel that the authors focused on is its ability to inhibit bacterial growth. On the other hand, the manuscript has also been well prepared. However, from a literature search it can be seen that this work was also reported in 2021 (https://doi.org/10.1016/j.cclet.2022.108069) but unfortunately it was not cited in this manuscript. Is it intentional or does it need explanation? Also, the author needs to explain the significance of the difference in data and the significant improvement from papers that have been published and those that are currently submitted.

Response: Thanks very much for promptly pointing out our errors. We had previously cited the mentioned paper in the original manuscript, but it may have been an oversight not to show it. Yes, this is continuous work and should be differentiated. We z previously only found the possibility of ZW having a gel but did not know that it was antimicrobial and did not consider using non-toxic and non-hazardous GDL for conditioning. Moreover, we revisited it in the introduction. The corrected part is on Page 2, Lines 21-25 of the main manuscript.

Comment 2: Rules for writing scientific names of microbial isolates to be considered.

Response: Thanks for your important suggestion. We have rechecked the presentation regarding the biological aspects. Based on your comments, we have redescribed the step-by-step experimental procedure in the antimicrobial experiments.The corrected part is on Page 7, Lines 1-25 of the main manuscript.

Comment 3: It has been said that the synthesized hydrogels have great potential for the treatment of skin or wound infections caused by Gram positive and Gram negative bacteria. This conclusion must be supported by data showing that the antibacterial effect of the hydrogel has a broad antibacterial spectrum. If the data already exists, it should be mentioned in this manuscript.  

Response: Thank you very much for this very important reminder. We have rechecked the conclusion section of the manuscript and found that there is indeed a problem, as you have described. Indeed, we only performed antimicrobial experiments in this study and did not perform additional organism testing. The previous statement lacked the necessary experimental data to support it. After referring to other published articles in this journal, we have made significant changes to the conclusions and added literature citations. Thanks again for your suggestions. The corrected part is on Page 10, Lines 5-14 of the main manuscript.

Comment 4: It is advisable that the introductory article of this manuscript is also cited.  

Response: Thanks for your important suggestion. After checking the complete text, we confirmed that the necessary literature had been cited and detailed descriptions had been made. The revised manuscript is more rigorous and careful. Thanks again for your suggestion. The corrected part is on Page 2, Lines 5-10 of the main manuscript.

Reviewer 2 Report

Abstract:  The abstract contains more general information. Mentioned the key findings and it is suggested to modified the abstract section

Authors should give informative objectives at the end of introduction section and discussed with recent works.

Please polish all the figures for a better presentation of the whole manuscript. The text is a blur in a few figures and makes color changes where appropriate.

Please provide the reaction mechanism in schematic form

The conclusion should be rephrased to be more concise and contain an overall conclusion.

Few references are very older, authors should replace it with recent references.

Author Response

Point-by-point response to the individual questions of Reviewer 2:

Over all Comment:

Comment 1: The abstract contains more general information. Mentioned the key findings and it is suggested to modified the abstract section

Response: Thank you for your reminder. Following your suggestion, we have reworked the abstract section of the manuscript, omitting the original abbreviations. Indeed, this will make the abstract more concise and readable.The corrected part is on Page 1, Lines 5 of the main manuscript.

Comment 2: Authors should give informative objectives at the end of introduction section and discussed with recent works.

Response: Thank you very much for your careful reading. We have revised the preface section of the manuscript following your comments and have added the necessary discussion and recent research advances. The corrected part is on Page 1-2 of the main manuscript.

Comment 3: Please polish all the figures for a better presentation of the whole manuscript. The text is a blur in a few figures and makes color changes where appropriate.  

Response: Thank you very much for your valuable suggestions. After revisiting the original manuscript, we rearranged the graphs of the paper. The figures have been brightened, and the formatting and figure sizes have been standardized, making it easier for the reader to understand the images' message. The revised article will be more conducive to readers' reading.

Comment 4: Please provide the reaction mechanism in schematic form

Response: Thank you for your reminder. Following your comments, we have carefully referred to other papers and revised the schematic. We hope that the revised content can be accepted by a wide range of readers. Thanks again for your careful reading. he corrected part is on Page 4 of the main manuscript.

Comment 5: The conclusion should be rephrased to be more concise and contain an overall conclusion.

Response: Thank you for your valuable comments. We have rechecked the description of the conclusion and there are indeed certain issues. Based on your suggestions, we have made additional additions and changes to the content. The corrected part is on Page 10 of the supporting information.

Comment 6: Few references are very older, authors should replace it with recent references.

Response: Thank you very much for this very important reminder. Based on your suggestion, we have added relevant literature published in recent years. Yes, you are right. The papers cited in our original manuscript are indeed older. Due to the value of these classic papers, they are still retained in the revised manuscript. Thank you again for your suggestion, which is very helpful to improve the quality of the paper.

Reviewer 3 Report

The manuscript entitled "One-step construction of tryptophan-derived small molecule hydrogels for antibacterial materials", discussed using gluconolactone (GDL) to adjust the pH of the solution to induce rapid self-assembly of N-[(benzyloxy)carbonyl]-L-tryptophan (ZW) to form a three-dimensional (3D) gel network, then developed a stable and effective small-molecule hydrogel with antibacterial properties.

The manuscript is well-organized and clearly stated. I would suggest accepting it after the following minor concerns are addressed. The detailed comments are shown below:

1. The background section is poorly presented and does not facilitate the reader's understanding of the background of the material study.

2. The figure 2 caption should be consistent with the previous section.

3. For figure 5, the markings in the picture does not match the figure caption.

4. In general, the unit format should be uniform throughout the text. The unit formats in the text are two including "cm-1" and "m2/g". Please standardize the unit format.

5. It must be ensured that the strain amplitude employed in the frequency sweep test is in the linear regime. Provide the strain sweep result to confirm that the frequency sweep was carried out in linear regim.

Author Response

Point-by-point response to the individual questions of Reviewer 3:

Over all Comment:

The manuscript is well-organized and clearly stated. I would suggest accepting it after the following minor concerns are addressed. The detailed comments are shown below:

Comment 1: The background section is poorly presented and does not facilitate the reader's understanding of the background of the material study.

Response: Thank you for your valuable comments. We have rechecked the description of the background and there are indeed certain issues. Based on your suggestions, we have made additional additions and changes to the content. The corrected part is on Page 1 of the supporting information.

Comment 2: The authors should pay a lot attention to accurate scientific presentation. The figure 2 caption should be consistent with the previous section.

Response: Thank you for pointing out our inadequacies. We're sorry for this low-level error in the original manuscript. This problem has been fully corrected in the revised manuscript after re-examination of the content and images. The modified pictures also include Figure 2. The corrected part is on Page 4 of the main manuscript.

Comment 3:  For figure 5, the markings in the picture does not match the figure caption.

Response: Thank you for pointing out our inadequacies. We apologize for not noticing these details in the original draft. The annotated part of Figure 5 has been redescribed in the revised manuscript. The corrected part is on Page 7 of the main manuscript.

Comment 4: In general, the unit format should be uniform throughout the text. The unit formats in the text are two including "cm-1" and "m2/g". Please standardize the unit format.

Response: Thanks for your careful reading. We have reunified these notations in the revised manuscript, such as "CFU/mL". In the literature, using "cm-1" for wavelength is an accepted notation for infrared spectra. We have retained this usage for the benefit of a wider audience. After carefully examining the entire text, these d-unit symbols and language expressions have been further improved. Thanks again for your careful reading, which is very important to improve the quality of our articles.

Comment 5: It must be ensured that the strain amplitude employed in the frequency sweep test is in the linear regime. Provide the strain sweep result to confirm that the frequency sweep was carried out in linear regim.

Response: We apologize for making a low-level mistake in the original draft. In our experiments, the frequency test should be in the range of 1 -100 rad/s, corresponding to the original graph. For the strain test, the frequency is actually fixed at 1 rad/s. From Figure 1, we can see that the modulus of the gel is in the linear region in the range of 1-10 rad/s. We have corrected this in the revised version and clearly marked it. The corrected part is on Page 3, Lines 14 of the main manuscript.